# Exercise-Based Interventions to Enhance Long-Term Sustainability of Physical Activity in Older Adults: A Systematic Review and Meta-Analysis of Randomized Clinical Trials

**DOI:** 10.3390/ijerph16142527

**Published:** 2019-07-15

**Authors:** Oriol Sansano-Nadal, Maria Giné-Garriga, Jennifer S. Brach, David M. Wert, Javier Jerez-Roig, Myriam Guerra-Balic, Guillermo Oviedo, Jesús Fortuño, Natàlia Gómara-Toldrà, Luis Soto-Bagaria, Laura Mónica Pérez, Marco Inzitari, Ivan Solà, Carme Martin-Borràs, Marta Roqué

**Affiliations:** 1Department of Physical Activity and Sport Sciences, Faculty of Psychology, Education and Sport Sciences (FPCEE) Blanquerna, Ramon Llull University, Císter 34, 08022 Barcelona, Spain; 2School of Health and Life Sciences, Glasgow Caledonian University, Cowcaddens Road, Glasgow G4 0BA, UK; 3Department of Physical Therapy, Faculty of Health Sciences (FCS) Blanquerna, Ramon Llull University, Padilla 326-332, 08025 Barcelona, Spain; 4Department of Physical Therapy, School of Health and Rehabilitation Sciences, University of Pittsburgh, Forbes Tower, 3600 Atwood St, Pittsburgh, PA 15260, USA; 5Research group on Methodology, Methods, Models and Outcome of Health and Social Sciences (M3O), Faculty of Health Science and Welfare, University of Vic-Central University of Catalonia (UVIC-UCC), Sagrada Família 7, 08500 Vic, Spain; 6Department of Health Sciences, Faculty of Health Sciences and Welfare, University of Vic, Sagrada Família 7, 08500 Vic, Spain; 7Intermediate Care Hospital, Parc Sanitari Pere Virgili, Avinguda de Vallcarca 169-205, 08023 Barcelona, Spain; 8RE-FiT Barcelona Research Group, Vall d’Hebrón Institute of Research (VHIR), Passeig de la Vall d’Hebron 119-129, 08035 Barcelona, Spain; 9Department of Medicine, Universitat Autònoma de Barcelona, Plaça Cívica, 08193 Bellaterra, Barcelona, Spain; 10Iberoamerican Cochrane Centre, Institute of Biomedical Research Sant Pau (IIB-Sant Pau), Sant Antoni Marià Claret 167, pavilion 18, 08025 Barcelona, Spain; 11CIBER Epidemiology and Public Health, CIBERESP, 28029 Madrid, Spain; 12Institute on Health and Aging, Universitat Autònoma de Barcelona (UAB), Sant Antoni Marià Claret 171 (Casa Convalecencia), 08041 Barcelona, Spain

**Keywords:** Older adults, physical activity, sustainability, adherence, meta-analysis, systematic review

## Abstract

Exercise is a form of physical activity (PA). PA is an important marker of health and quality of life in older adults. The purpose of this study was to conduct a systematic review of the literature to assess the effect of exercise-based interventions on an at least six-month follow up PA measure, and to describe the specific strategies implemented during the intervention to strengthen the sustainability of PA in community-dwelling 65+ year-old adults. We registered and conducted a systematic review and meta-analysis (PROSPERO: CRD42017070892) of randomized clinical trials (RCT). We searched three electronic databases during January 2018 to identify RCT assessing any type of exercise-based intervention. Studies had to report a pre-, post-, and at least 6-month post-intervention follow-up. To be included, at least one PA outcome had to be assessed. The effect of exercise-based interventions was assessed compared to active (e.g., a low-intensity type of exercise, such as stretching or toning activities) and non-active (e.g., usual care) control interventions at several time points. Secondary analyses were conducted, restricted to studies that reported specific strategies to enhance the sustainability of PA. The intervention effect was measured on self-reported and objective measures of time spent in PA, by means of standardized mean differences. Standardized mean differences of PA level were pooled. Pooled estimates of effect were computed with the DerSimonian–Laird method, applying a random effects model. The risk of bias was also assessed. We included 12 studies, comparing 18 exercise intervention groups to four active and nine non-active control groups. Nine studies reported specific strategies to enhance the long-term sustainability of PA. The strategies were mostly related to the self-efficacy, self-control, and behavior capability principles based on the social cognitive theory. Exercise interventions compared to active control showed inconclusive and heterogeneous results. When compared to non-active control, exercise interventions improved PA time at the six-months follow up (standardized mean difference (SMD) 0.30; 95%CI 0.15 to 0.44; four studies; 724 participants; I^2^ 0%), but not at the one- or two-years follow-ups. No data were available on the mid- and long-term effect of adding strategies to enhance the sustainability of PA. Exercise interventions have small clinical benefits on PA levels in community-dwelling older adults, with a decline in the observed improvement after six months of the intervention cessation.

## 1. Introduction

Many health-oriented systems and organizations are faced with the challenge of implementing new health-related practices, and while many of these programs demonstrate initial success, they fail to become a habit or routine for the participants. Policy makers and other stakeholders are increasingly concerned with the long-term impact of their investment. Greenhalgh et al. [1] pointed out there was a near absence of studies focusing primarily on how to sustain health promotion interventions in the long term.

While intervention sustainability is defined as the continued use of intervention components and activities for the continued achievement of desirable health outcomes within the population of interest [2], determining how to foster the maintenance of health-related benefits acquired by the intervention’s recipients once it ends is another great challenge [3]. Adherence, similarly, is defined as the extent to which a person’s behavior (e.g., lifestyle changes) corresponds with agreed recommendations from a health care provider [4], based on completion (e.g., retention), attendance, duration, and intensity while undergoing a certain intervention.

The incidence rate of many chronic diseases increases with age [5,6]. Physical activity (PA) is a major aspect of chronic disease self-management [7,8,9,10,11,12,13], and higher levels of PA are associated with healthy ageing [14]. In particular, for older adults, balance and resistance training programs are an effective way to maintain mobility and independence [9]. People who undertake regular PA tend to experience better health and live longer [15,16], even with small increases each day [16]. However, insufficient amounts of PA remain one of the major behavioral burdens worldwide [17,18,19]. Older adults are the less active group, with only about 11% meeting the current PA recommendations [20,21], creating a ripe environment for PA intervention research. The road towards sustainable maintenance to PA might be the cornerstone of health promotion.

Interventions to promote maintenance of PA once the intervention ends in older adults have achieved limited success, particularly over the long term [14,22,23,24,25,26,27]. A clear definition of long-term sustainability is still lacking, so we will consider long-term as a person maintaining a recommended behavior over a 12-month period [4]. Most current studies thus far have focused on identifying the factors that are critical to the success of initial implementation efforts. The development of alternative approaches based on end-users’ preferences, and which implement behavioral change concepts, have been repeatedly requested [28]. Some of the most used theories in health behavior change include psychological theories, such as the transtheoretical model [29] and the social cognitive theory (SCT) [30]. Behavioral change techniques (BCTs) include cognitive and behavioral processes of change, and had been associated with an increase in the effects of behavior change interventions [31]. There is a need for designing effective and sustainable interventions to promote PA in the long term in older adults [3].

The purpose of this study was to conduct a systematic review of the literature to assess the effect of exercise-based interventions, implemented in community-dwelling older adults (65+ years of age), on an at least six-month follow up of PA—and subsequently describe the specific strategies implemented to strengthen the long-term sustainability of PA. We hypothesized that interventions that described specific strategies to enhance the sustainability of PA practice would be more effective in PA maintenance, at least after the six-month post-intervention cessation. Our paper aimed to inform an agenda for research, funding, and policies on PA promotion in older adults.

## 2. Materials and Methods

### 2.1. Study Selection

We conducted a systematic review of the literature to assess the effect of exercise-based interventions, implemented in community-dwelling older adults, at least six-months post-intervention, on the sustainability of PA (PROSPERO: CRD42017070892). The guidelines for conduct and a report of systematic reviews in the Cochrane Handbook for Systematic Reviews of Interventions Version 5.1.0 [32] and PRISMA statement [33] were followed. We searched MEDLINE, EMBASE, and CENTRAL up to January 2018 without language or date restrictions. We designed a search strategy adapted to the requirements of each database, combining their controlled vocabularies and text terms. We used, among others, keywords like older, elderly, ageing, aging, sustainable, sustainability, maintenance, adherence, long-term effects, and physical activity, exercise, rehabilitation (see Appendix A). We approached other sources (references from relevant systematic reviews and meta-analyses, or personal communication with experts) to identify additional studies. Title words had to include ‘exercise’ or ‘physical activity’ or ‘exercise referral schemes’ and ‘older adults’. Results from the search procedures were imported into Mendeley bibliographic software and duplicates were removed. Study titles and abstracts of identified studies were screened by pairs of authors independently (M.G.-G., N.G.-T., M.G.-B., L.S.-B., J.J.-R, O.S.-N., G.O., J.F., J.S.B., D.M.W.) to exclude studies that clearly did not meet the inclusion criteria. Full text versions of potentially eligible studies were retrieved and assessed independently by pairs of authors (M.G.-G., N.G.-T., J.S.B., D.M.W., L.S.-B., L.M.P., M.I., M.G.-B., J.J.-R., O.S.-N. G.O., J.F.) against the inclusion criteria, and agreements were reached by consensus. Articles were excluded at all stages of the review and the reasons for exclusion were recorded (see Figure 1: study flow chart).

Studies were included if they were randomized clinical trials (RCT) involving any type of exercise program (e.g., exercise referral scheme, aerobic and/or strength exercise programs, tai-chi) in community-dwelling older adults aged 65 years or older of both genders. Studies had to report at least pre-, post-, and at least six-month post-intervention follow-up exercise intervention measurements. Furthermore, at least one valid PA outcome had to be assessed (e.g., self-report, activity monitor) as a primary or secondary outcome measure. Studies were included if the exercise-based intervention was compared to either a non-active control, such as usual care, or an active control, if participants performed a low intensity type of exercise such as stretching or toning activities, or physiotherapy. We included studies aimed at assessing the sustainability of PA and studies that assessed PA as a health-related outcome measure.

Studies with participants younger than 65 years old were only included in the review if the mean age was over 65, and the study either mostly included participants over 65, or presented results specifically for the subgroup of 65+ participants.

### 2.2. Quality Assessment

The validity of the results provided by the studies was assessed using the Cochrane Collaboration’s risk of bias tool [32] by two independent reviewers, with two added questions. The domains of selection bias, performance bias, outcome assessment bias, attrition bias, and selective reporting bias were assessed through the following measures: (1) random sequence generation; (2) allocation concealment; (3) adherence to intervention; (4) contamination; (5) objective PA assessment; (6) incomplete outcome data; and (7) incomplete or selective reporting, based on published papers and documentation in trial registries. The risk of bias for each item could be rated as ‘high’ (+), ‘low’ (−), or ‘unclear’ (?).

Performance bias items related to the blinding of participants and personnel were not included, as this is typically not feasible for exercise interventions. The assessments of both reviewers were compared, and disagreements were resolved by discussion (see Appendix A).

### 2.3. Data Extraction

Pairs of independent reviewers (M.G.-G and N.G.-T., J.S.B. and D.M.W., L.S.-B. and L.M.P., M.I. and M.G.-B., J.J.-R. and O.S.-N., G.O. and J.F.) extracted information about the mean age, inclusion and exclusion criteria, setting, PA and physical function outcomes, instruments, and time point measured (Appendix A). The extraction of specific details on intervention components included dose (frequency, intensity, time, and type), profile of the professional delivering it, control intervention description, compliance, and the specific strategies targeted to increase the sustainability of PA practices and its relevant theory background (Appendix A).

Data regarding study characteristics was extracted into an electronic database. Three independent reviewers checked the accuracy of this procedure (M.G.-G., M.R., C.M.-B.) and if any discrepancy arose, consensus through discussion was sought.

### 2.4. Outcomes

The primary outcome of the review was time spent in PA, either assessed with valid objective measures with any kind of activity monitor (e.g., accelerometer or pedometer) or self-reported instruments. Information on the instruments used in each study (e.g., validation and recall periods of the self-reported measures) can be found in Appendix A.

### 2.5. Statistical Analysis

We assessed the effect of exercise-based interventions on PA levels at several time-points (immediately after intervention, at mid-term (between 6 and 11 months after), and long-term (12 or more months after)). Data at three-months follow-up from a study [34] have been included in the mid-term category for comprehensiveness purposes.

The effect of exercise-based interventions was organized in two comparisons, defined by the type of control: exercise-based interventions compared to active control interventions (e.g., a low-intensity type of exercise, such as stretching or toning activities, or physiotherapy); and compared to a non-active control (e.g., usual care). We conducted secondary analyses restricted to those studies that reported specific strategies to enhance the sustainability of PA.

The effect of intervention on objective and self-reported time spent in PA was assessed with standardized mean differences, due to the variability of PA measures applied across trials. Pooled estimates of effect were computed with the DerSimonian–Laird method, applying a random effects model. Heterogeneity was assessed by means of the I^2^ statistic, considering values over 50% to indicate serious inconsistency [32]. When high statistical or clinical heterogeneity was found, the pooled estimates of effect were unreliable.

Three studies comparing either two or three active interventions to one control arm [35,36,37] were included in the meta-analysis by merging their intervention arms into a single arm, to avoid double counting of data. Data from a four-arm study [38] was presented as two comparisons.

## 3. Results

### 3.1. Description of Study Inclusion (Figure 1)

The search update identified 2270 records, of which 46 were eligible after title and abstract screening. After the full-text assessment, 15 studies [34,35,36,37,38,39,40,41,42,43,44,45,46,47,48] remained eligible for inclusion, but three studies were later excluded [40,46,47], one for reporting the same PA measures and population as another included study [40], and two for not having a real randomized control group and/or not having a six-month follow up assessment [46,47]. The 12 studies that were included in the final analyses investigated a total of 18 exercise-based arms [34,35,36,37,38,39,41,42,43,44,45,48], compared to four active control groups [34,36,42,44] and nine non-active control groups (usual care) [35,37,38,39,41,43,45,48]. It is worth noting that the targeted older adults were heterogeneous in terms of comorbidities and functional status.

The principal investigators of four studies were contacted because of incomplete reporting of PA data [35,38,41,42]. Baseline data of one included study [38] were retrieved in a previous article [49]. After personal communication, we obtained additional information for two studies [35,38].

### 3.2. Characteristics of Exercise Arms Included in the Meta-Analysis

We described the exercise arms assessed in the included trials in Appendix A, and the description of the interventions and strategies applied to enhance long-term sustainability of PA levels is presented in Appendix A. Trials included a range of 52 to 422 participants, that were, on average, 75.8 years old and 73.7% were female. Only three trials reported specific information regarding the race/ethnicity of the participants, with between the 80% and 95% of the sample being Caucasian [36,42,44].

Intervention duration varied across the 12 studies, ranging from 8 weeks to 24 months. All interventions were supervised, the frequency of supervised exercise sessions was mainly two or three times a week, and session duration ranged from 30 to 90 minutes. Prescribed exercise intensity was not described in most of the included studies [36,37,39,41,42,43,45] (Appendix A).

### 3.3. Comparison 1: Exercise-Based Intervention Versus Active Control

Three studies with 265 participants compared active interventions against active controls. Two of the studies assessed intervention effect on self-reported measures of PA [36,44], while the third study applied both self-reported and objective measures of PA [34]. Two of the studies implemented strategies to enhance sustainability [36,44] and one study [34] did not. The main analyses showed heterogeneity in self-reported estimates of PA, at all-time points. One of the studies showed a consistent effect of the intervention immediately and at the six-month and two-year follow ups [44], while the other two showed either no effect [36], or an initial immediate effect that disappeared during follow ups [34] (Figure 2). The secondary analyses restricted to the two trials with sustainability-enhancing strategies reached similar results and heterogeneity levels as the main analyses (Appendix A). The only study measuring PA objectively [36] showed no differences between intervention and active control, neither immediately post-intervention (SMD 0.16; 95%CI −0.30 to 0.62; 100 participants) nor at the six-month follow up (SMD 0.15; 95%CI −0.31 to 0.61).

### 3.4. Comparison 2: Exercise-Based Intervention versus Non-Active Control

Eight studies with 1676 participants compared exercise-based interventions against non-active controls [35,37,38,39,41,43,45,48]. All but two studies implemented sustainability-enhancing strategies [35,37]. Five studies provided only on self-reported PA data [35,39,41,45,48], one study provided only objective PA data [43], and one study provided both types of data [38]. One additional study could not be included in the analysis for not presenting results data for the intervention and control groups separately [42].

Active interventions had a small effect on self-reported PA time compared to non-active control, both immediately post intervention (SMD 0.18; 95%CI −0.01 to 0.37; five studies [35,38,39,45,48]; 1257 participants; I^2^ 63%), and at the six-month follow up (SMD 0.30; 95%CI 0.15 to 0.44; four studies [39,41,45,48]; 724 participants; I^2^ 0%). The effect was gradually lost over time, with irrelevant long term results at the one-year follow up (SMD 0.27; 95%CI 0.05 to 0.48; one study [48]; 239 participants) and the two-year follow up (SMD 0.03; 95%CI −0.18 to 0.24; one study [38]; 350 participants) (Figure 3). When restricting the analyses to the four studies who applied some kind of strategy to enhance sustainability, the immediate post-intervention effect became significant and less heterogeneous (SMD 0.25; 95%CI 10.0 to 40.0; four studies [38,39,45,48]; 1108 participants; I^2^ 31%) (Appendix A).

The three studies which provided data with objective measures of PA were extremely heterogeneous, with two of them showing no differences between intervention and non-active control, either immediately or at any time point [36,37], and the third study showing significant differences in the changes between the baseline and the nine-month follow up [43] (Figure 4). The secondary analysis restricted to two studies which applied sustainability-enhancing strategies was still quite heterogeneous (Appendix A).

### 3.5. Reported Strategies to Enhance Long-Term Sustainable Maintenance of PA

In the present review, we classified the strategies according to the SCT (e.g., self-efficacy, behavioral capability, reinforcements, observational learning, expectations, expectancies, and self-control) [30]. Nine studies reported specific strategies to enhance the long-term sustainability of PA once the intervention had ended [36,38,39,41,42,43,44,45,48]. In one study, the authors offered for the intervention group to continue the PA sessions once a week, and informed the control group about a training program available in the community [39]. Similarly, in one study, researchers invited the participants to keep their PA monitors after the study, to favor sustaining the PA practice [36]. In another study, the person delivering the intervention applied the principles of self-efficacy, regular performance feedback, and positive reinforcement to enhance the motivation for exercise progression and maintenance [41]. McAuley et al. used self-efficacy as the guiding theoretical construct, and participants were sent feedback on their measures by mail as a reinforcement strategy, so that they could see their improvements in health outcomes [42]. In two studies, patients kept an activity diary and agreed to establish weekly goals for increased activity with the training provider [43,45]. In another study, each participant engaged in weekly group-mediated behavioral counselling sessions that focused on self-regulatory skills central to promoting PA, for the first 10 weeks [44]. In the study that assessed outcomes two years post-intervention, participants were encouraged to keep their pedometers and were invited to continue with the exercise training, or encouraged to participate in any other preferred PA [38]. In the most recent study, the exercise specialist identified a leader in each exercise group to organize a third session each week on their own, to enhance the autonomy of the participants and the sustainable maintenance of PA practice [48]. All participants were offered a personalized exercise program and visits with all participants were made to the nearest community resources (e.g., sport facilities) [48]. This latest intervention included mechanisms to enhance social support during the cool-down phase of each session, such as social influence/social comparison, social control, self-esteem, sense of control and belonging, and companionship [48].

The sustainability-enhancing strategies most associated with success were two behavioral strategies: self-control and behavioral capability. Self-control strategies used by participants to maintain self-motivation and achieve personal goals were the use of tracking monitors, daily step diaries, and individualized step-based goals. Behavioral capability strategies included individualized counselling on the role of PA in disability prevention, designed to promote PA, and visits to community resources (e.g., sport facilities) where the regular PA practice could be continued.

## 4. Discussion

The current systematic review synthesized the data from 12 RCTs, including 18 exercise-based arms, compared to four active control and nine non-active control groups that evaluated the effects of exercise-based interventions implemented in older adults to sustain the maintenance of PA increases for the mid- and long-term, and to describe the strategies used in the interventions to enhance the sustainability of PA in this population.

Results of the effect of exercise interventions on the sustainability of PA were heterogeneous in the present study. The results of a recent meta-analysis also revealed a high heterogeneity associated with the pooled treatment, with a significant small effect favoring exercise interventions over control [25,26]. As the authors included non-randomized trials, it is possible that the effect reported was overestimated [25,26]. Supervised exercise interventions with specific strategies to enhance sustainability had statistically significant beneficial effects on PA levels at the end of intervention compared to non-active controls or active controls. However, at 12 months post-intervention cessation, benefits subsided. Unclear effects on maintenance beyond 12 months were noted in other reviews of PA promotion, due to a lack of high-quality longitudinal studies, heterogeneity of interventions, and a high chance of bias [27].

Researchers tended to describe some strategies to enhance PA sustainability in the intervention, although most researchers failed to design the intervention bearing sustainability in mind [3]. Approximately 80 theories related to behavior change had been identified in the literature, and one of the most used is the SCT [50]. The specific strategies described to enhance sustainability in the included studies might not be applicable to all populations, or might not be explicitly based on theory. Choosing a relevant theory can be a challenging task for intervention designers, especially given the large number of theories.

### 4.1. Mechanisms that Underpin our Main Finding

Insufficient PA is a modifiable lifestyle factor that leads to an increasing pressure to deal with additional health care costs associated with an ageing population and chronic disease burden [18]. A recent systematic review suggested that participants who engaged in PA had their odds of living a healthy life in older age increased compared to participants that were physically less active [14]. The present systematic review does not show a successful picture of exercise-based interventions to increase PA levels in older adults in the long-term, as only one intervention that added specific strategies to sustain PA maintenance showed better effects in the long-term [43]. Public health interventions targeting insufficient PA levels are usually developed without the involvement of end-users, which seems to be the case for all the studies included in the present review. One promising way to develop effective PA-based interventions (e.g., exercise-based) is to combine theory-based with solution-based approaches, characterized by considering individuals’ interests, forming cooperative teams of stakeholders, and distributing actions and decisions [51]. Patient and public involvement (PPI) has developed into an integral part of research practice over the last 25 years. It is thought that involving end-users in the development of solution-based interventions using key elements derived from participatory methodologies such as PPI may increase the likelihood of producing sustainable change [52]. However, none of the studies included in our review reported any approach of PPI in the design of the exercise-based intervention.

Behavioral-type components, such as self-monitoring, goal setting, and prompting, engage participants in actively changing physical behaviors [53]. Cognitive-type components, such as education, problem solving, or counselling, promote change in cognitive processes, attitudes, or beliefs [54]. Some of the most cited strategies used in the included studies were education [55], self-monitoring [56], and action planning [57]. These have all been acknowledged in the literature as important theoretical constructs for successful behavior change, however, in the included studies they might not have been sufficiently addressed in the intervention design and across the intervention period [50]. A recent narrative systematic review supports combination strategies as more successful in changing PA behavior [58] than interventions that target a standalone behavioral strategy. However, none of the strategies described in the present review suggested the inclusion of the end-user themselves, which may be one of the most vital components for finding the right motivating factors to improve long-term exercise/PA participation among older adults. The behavioral capability strategies reported in some of the included studies aimed at engaging participants in existing community resources. Community-based resources to facilitate health behavior change may include a wide range of available opportunities, such as city-based exercise programs [59], referral to fitness professionals, or fitness centers [60]. Referrals to specific community programs, such as exercise for adults, have shown a positive effect on patient behavior [60,61]. Self-control strategies are aimed at providing regular performance feedback to enhance the motivation for activity maintenance (e.g., pedometer, daily steps diary). Several studies have shown the positive effects of activity trackers for enhancing PA levels. However, while some short-term studies support success, more recent data question the longer-term effects on PA change [62,63].

It is important to note that this area of PA research is characterized by a dearth of knowledge of the effects of exercise interventions on long-term sustainability, particularly among older adults [64]. Brawley et al. highlighted the need for longitudinal research investigating not only the factors associated with activity maintenance after the program ends, but also the psychological strategies that were most effective for attenuating the rate of declining activity levels beyond program completion [64]. A whole system-oriented approach is required that is tailored to meet the needs of older adults and aligns with social, individual, and environmental factors [31]. A patient-centered approach supporting clinical reasoning to detect needs, limitations, and strengths in both the participant and the physical and social environment and choosing evidence-based interventions to enhance behavior change was described in one study, which had to be excluded from the present review for not reporting a six-month follow-up [47]. Mechanisms to enhance social support and participation might be effective for battling social isolation and enhancing adherence [48].

PA is continuously ubiquitous throughout the day [65], so understanding usual PA patterns might help turn daily routines into opportunities for exercising rather than performing specific exercise programs [66]. Interventions should be developed in coherence with their daily routine, tailored to an individual’s context and circumstances to improve adherence [67].

### 4.2. Strengths and Limitations

To our knowledge, this is the first systematic review focusing on the sustainability of exercise-based interventions to maintain increases in PA levels in older adults, in the mid- and long-term. The strengths of the current meta-analysis are the extensive search strategy, the inclusion of RCTs, and the number of exercise arms, as well as the description of specific strategies to enhance sustainability of PA practice. However, the present systematic review is restricted to three databases. In addition, two studies could not be included due to not being able to contact the main author. The meta-analyses were performed on few studies, which caused estimates to be more imprecise. This imprecision, added to the observed heterogeneity, limits our confidence in the results. On the other hand, even though we aimed at describing the long-term effects of current interventions on enhancing PA sustainability, very few studies reported this, so the analysis had to focus on the six-month follow up period.

Besides this, most of the studies included in the review assessed PA with a self-report instrument, increasing the impact of recall bias [68]. There was also lack of consistency regarding the way frequency, intensity, and duration of PA were reported. Similar problems are also mentioned in other systematic reviews of PA [14,24,26,28].

## 5. Conclusions

Exercise interventions are statistically significant but translate to a small clinical benefit on PA levels in community-dwelling older adults. Among those studies that reported longer-term follow up (six months after the intervention cessation) improvements have been shown to decline. Only one intervention that added specific strategies to enhance sustainability demonstrated maintenance of PA practice in the long term. Further research is needed to conclude the most effective strategies to enhance the sustainability of PA for older adults.

This review could be of interest to clinical or institutional leaders and administrators, as well as to health professionals and researchers in order to inform clinical decisions and policies.

## Data Statement

The data will be made available to interested researchers. Please contact the corresponding author.

## Figures and Tables

**Figure 1 ijerph-16-02527-f001:**
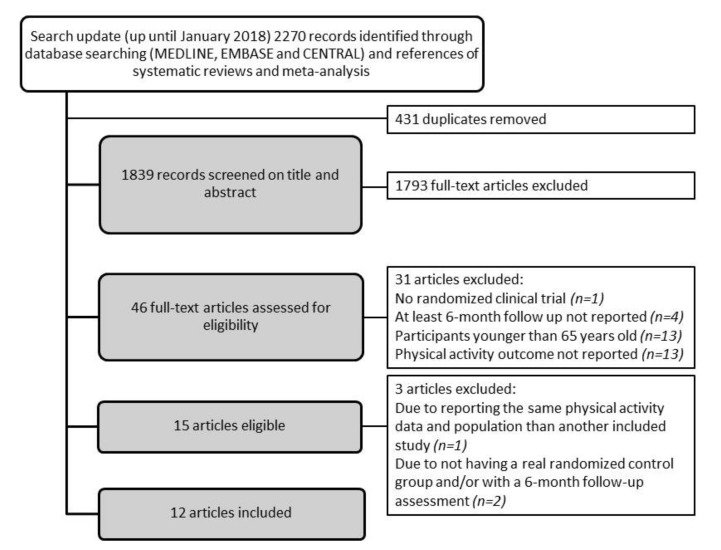
Flow chart of study inclusion.

**Figure 2 ijerph-16-02527-f002:**
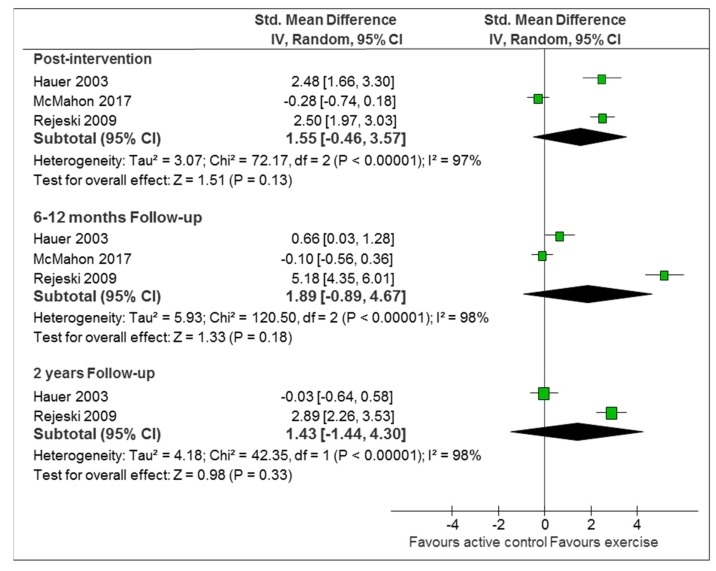
Exercise-based intervention versus active control (self-reported PA: physical activity). Abbreviations: Std. Mean Difference: standardized mean difference; IV, Random: a random-effects meta-analysis is applied, with weights based on inverse variances; 95% CI: 95% confidence interval; df: degrees of freedom; Tau^2^ and I^2^: heterogeneity statistics; Chi^2^: the chi-squared test value; Z: Z-value for test of the overall effect; P: p value.

**Figure 3 ijerph-16-02527-f003:**
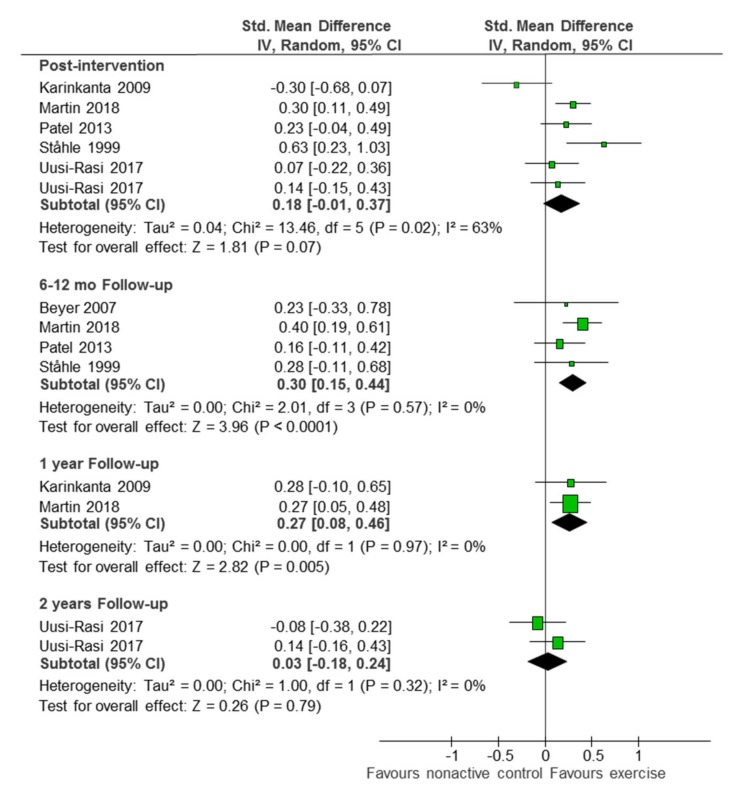
Exercise-based intervention versus non-active control (self-reported PA). Abbreviations: Std. Mean Difference: standardized mean difference; IV, Random: a random-effects meta-analysis is applied, with weights based on inverse variances; 95% CI: 95% confidence interval; df: degrees of freedom; Tau^2^ and I^2^: heterogeneity statistics; Chi^2^: the chi-squared test value; Z: Z-value for test of the overall effect; P: p value.

**Figure 4 ijerph-16-02527-f004:**
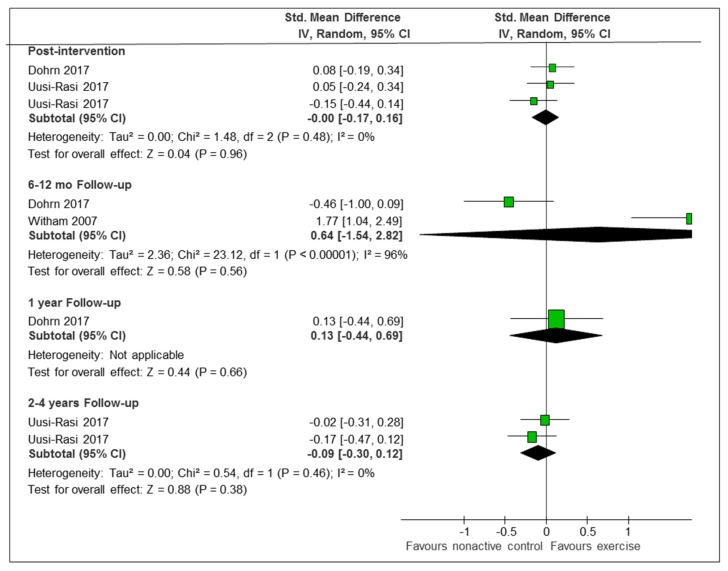
Exercise-based intervention versus non-active control (objective measures of PA). Abbreviations: Std. Mean Difference: standardized mean difference; IV, Random: a random-effects meta-analysis is applied, with weights based on inverse variances; 95% CI: 95% confidence interval; df: degrees of freedom; Tau^2^ and I^2^: heterogeneity statistics; Chi^2^: the chi-squared test value; Z: Z-value for test of the overall effect; P: p value.

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
