# Peer review of "Exercise-Based Interventions to Enhance Long-Term Sustainability of Physical Activity in Older Adults: A Systematic Review and Meta-Analysis of Randomized Clinical Trials"

_ijerph, 2019, doi:10.3390/ijerph16142527_

Round 1

Reviewer 1 Report

In this study, the authors conducted a systematic review of the literature, with two purposes – to examine whether PA was continued at least 6-months after an exercise intervention for community dwelling older adults concluded and to describe the strategies used to improve PA adherence post-intervention. The latter purpose is especially important and should be the primary focus of this review. Overall, the study itself was well written and there were several interesting findings. My main criticism is that I would like to see more detail and focus on the strategies that were found to be at least somewhat successful. My comments by section:

Materials and Methods:

Section 2.2: Quality Assessment paragraph (lines 149-150): Consider rewriting this sentence.

Section 2.4: Outcomes paragraph (lines 164-166): It should be indicated whether the self-reported measures were validated. Some self-report of PA measures have not been validated. Others (like the “Community Healthy Activities Model Program for Seniors”, CHAMPS) have.

Results:

Section 3.2: Include race/ethnicity of the participants. It may also be warranted to give consideration to the built environment or similar factors (urban, rural), if possible.

Sections 3.4 and 3.5 look at adherence when PA programs were compared with either an active control or standard care. This fits with the first purpose of the study. Section 3.3 fits with the second purpose of the study. It may be worthwhile to move section 3.3 to end of the results. Additionally, it would be very worthwhile is the authors can expand more upon which strategies were most associated with success (adherence), post-intervention rather than only state what strategies were used among the studies scoured.

It would also be interesting to see how post-program adherence compared between PA interventions that were strength, balance, aerobic, or some combination.

Funding: Funding sources (if any) were not disclosed in the manuscript. This section needs to be completed.

Author Response

REVIEWER 1

Dear reviewer,

Thank you very much for the time taken to review our systematic review. Please see our answers to your comments in italics, and the changes in the manuscript underlined in yellow. Don’t hesitate to contact us for further information.

In this study, the authors conducted a systematic review of the literature, with two purposes – to examine whether PA was continued at least 6-months after an exercise intervention for community dwelling older adults concluded and to describe the strategies used to improve PA adherence post-intervention. The latter purpose is especially important and should be the primary focus of this review. Overall, the study itself was well written and there were several interesting findings. My main criticism is that I would like to see more detail and focus on the strategies that were found to be at least somewhat successful.

Thank you very much for your encouraging comments. We have tried to strengthen the information regarding the strategies found to be more successful in enhancing PA adherence in page 9 (section 3.5) and page 10 (section 4.1) of the discussion.

Materials and Methods

Section 2.2: Quality Assessment paragraph (lines 149-150): Consider rewriting this sentence.

Thank you for the suggestion. This sentence has been deleted, and the previous one was modified: “Performance bias items related to blinding of participants and personnel were not included, as it is typically not feasible for exercise interventions”.

Section 2.4: Outcomes paragraph (lines 164-166): It should be indicated whether the self-reported measures were validated. Some self-report of PA measures have not been validated. Others (like the “Community Healthy Activities Model Program for Seniors”, CHAMPS) have.

As suggested, we have indicated whether the self-reported measures had been validated or not in Supplementary Table S3.  

Results

Section 3.2: Include race/ethnicity of the participants. It may also be warranted to give consideration to the built environment or similar factors (urban, rural), if possible.

Unfortunately, nine of the twelve included studies did not specifically report the race/ethnicity of the participants, only the setting where the study took place:

-        Ståhle et al. 1999: Not reported (Karolinska Hospital, Stockholm)

-        Hauer et al. 2003: Not reported (Postward rehabilitation in a geriatric hospital in Germany)

-        Beyer et al. 2007: Not reported (Hospital Research in Copenhagen)

-        McAuley et al. 2007: 95% White

-        Witham et al. 2007: Not reported (Patients were recruited from the local heart failure clinic and Medicine for the Elderly clinic)

-        Karinkanta et al., 2009: Not reported (Tampere, Finland)

-        Rejeski et al., 2009: 18.2% were African American (Winston-Salem, Carolina del Nord)

-        Patel et al., 2013: Not reported (Auckland, New Zealand)

-        Dohrn et al,. 2017: Not reported (Stockholm County, Sweden)

-        McMahon et al., 2017: 80% caucasian (Minneapolis, Minnesota)

-        Uusi-Rasi et al., 2017: Not reported (Tampere, Finland)

-        Martin-Borràs et al., 2018: Not reported (Catalonia, Spain)

We have included the following sentence in section 3.2: “Only three trials reported specific information regarding the race/ethnicity of the participants, being Caucasian between the 80 and 95% of the sample [35,41,43].

Sections 3.4 and 3.5 look at adherence when PA programs were compared with either an active control or standard care. This fits with the first purpose of the study. Section 3.3 fits with the second purpose of the study. It may be worthwhile to move section 3.3 to end of the results. Additionally, it would be very worthwhile if the authors can expand more upon which strategies were most associated with success (adherence), post-intervention rather than only state what strategies were used among the studies scoured.

We thank the reviewer for his/her comment. We have moved section 3.3 accordingly, and we have tried to strengthen the information regarding the strategies found to be more successful in enhancing adherence in page 9 (section 3.5) and page 10 (section 4.1) of the discussion.

It would also be interesting to see how post-program adherence compared between PA interventions that were strength, balance, aerobic, or some combination.

I'm afraid we were not able to understand what the reviewer meant with this question. We would be more than happy to answer it with further clarification. Adherence and compliance to the intervention of the studies that reported it can be found in Supplementary Table S4.

Funding: Funding sources (if any) were not disclosed in the manuscript. This section needs to be completed.

We apologize and have added the following sentence: “This research received no external funding”.

Reviewer 2 Report

Summary-

This manuscript describes a systematic review and meta-analysis of RCTs with exercise-based interventions and their impact on mid- and long-term physical activity levels in older adults. The manuscript also discusses the behaviour change theories implemented in a portion of the RCTs.

Broad comments-

The topic of this review is interesting and, as laid out in the introduction, it is a very important area in healthcare. However, there are many fundamental discrepancies surrounding the research question and methods carried out in this study.

The title and purpose are incongruent with the review that was conducted. ‘Long-term adherence to physical activity’ was included in the title and was part of the purpose, yet the study design did not address this. Adherence, as defined in the introduction (lines 79-81), includes an agreed recommendation to adhere to, yet most of the studies do not set out an explicit level of PA to adhere to, rather PA was measured as an outcome to track change. It could be argued that these exercise interventions improved or promoted higher levels of PA, but the participants were not adhering to a specific recommendation. Additionally, while the title states ‘long-term adherence’ most of the focus of the paper is on 6-month follow up which does not fit the long term criteria set out in the introduction: ‘we will consider long-term for a person to maintain a recommended behaviour over a 12-month period.’

Other comments-

Inclusion/Exclusion criteria:

·         How is age > 65 y.o. determined, for example if participants > 60 y.o. were included in a study? Would this study be excluded?

·         Were pilot or feasibility RCTs included?

·         Hauer 2003 does not appear to have a homogenous community dwelling participant population, was there a discussion around including this study? And on what grounds led to the decision to include it?

Statistical analysis:

·         The meta-analysis is performed on very few studies generally, as few as one study from the 1-year follow up in Figure 4, but on average only 2-3 study results were included (Figures 2-4). This could be discussed as a limitation of the meta-analysis.

Methods:

·         As the number of records screened (1839) seems like a manageable number in this type of review, could you explain the reasoning for the number of reviewers who performed the screening (10 for abstract screening and 12 for full text screening)? Can you explain the reasoning behind data extraction performed in groups (5 pairs and one group of 3)?

·         What was the mechanism for standardisation of reviewing for so many reviewers? Was there training for each reviewer using the screening tool?

·         At the end of the data extraction section 2.3, there is a discrepancy between the number of reviewers who checked the accuracy.

·         Search strategy: as it is common practice to have searches within the last 12 months, it appears that an updated search would be needed.

·         Additionally, the manuscript states the search was up to January 2018, however the Supplementary Table S1 show the searches conducted in October and November 2017.

Author Response

REVIEWER 2

Dear reviewer,

Thank you very much for the time taken to review our systematic review. Please see our answers to your comments in italics, and the changes in the manuscript underlined in yellow. Don’t hesitate to contact us for further information. The present manuscript has been reviewed and approved for English language and style by two North American native researchers.

The title and purpose are incongruent with the review that was conducted. ‘Long-term adherence to physical activity’ was included in the title and was part of the purpose, yet the study design did not address this. Adherence, as defined in the introduction (lines 79-81), includes an agreed recommendation to adhere to, yet most of the studies do not set out an explicit level of PA to adhere to, rather PA was measured as an outcome to track change. It could be argued that these exercise interventions improved or promoted higher levels of PA, but the participants were not adhering to a specific recommendation. Additionally, while the title states ‘long-term adherence’ most of the focus of the paper is on 6-month follow up which does not fit the long term criteria set out in the introduction: ‘we will consider long-term for a person to maintain a recommended behaviour over a 12-month period.’

The authors appreciate your comment and agree that most studies did not have adherence to PA as their main purposes, and assessed PA as a health-related outcome measure. When we decided the aim of our review we were mainly interested in the interventions aimed at enhancing adherence to PA but we decided not to narrow the search excluding studies that provided data on PA without being adherence the end point. We have tried to clarify this point in page 3 section 2.1.

Regarding the reviewer’s second issue, even though we wanted to focus on the long-term follow up (over 12-month period), very few studies reported it, so that we focused on the 6-month follow up period. We have added a sentence in the limitations section in page 10 section 4.2.  

Inclusion/Exclusion criteria

How is age > 65 years old determined, for example if participants > 60 years old were included in a study? Would this study be excluded?

The authors had tried to clarify this point. We included older adults 65 years or older based on the World Health Organization’s definition of older adults. However, we included studies that recruited younger participants, if the mean age of the included participants was over 65, and either the study mostly included participants over 65, or it presented results specifically for the subgroup of 65+ participants.

This has been clarified in the inclusion criteria section in page 3.

Were pilot or feasibility RCTs included?

Yes, we did not exclude pilot or feasibility studies.

Hauer 2003 does not appear to have a homogenous community dwelling participant population, was there a discussion around including this study? And on what grounds led to the decision to include it?

The authors decided to include Hauer’s study even though the participants had a history of injurious falls, and the setting was a geriatric hospital. We considered that the study accomplished the inclusion criteria and despite being a rather specific frailer population, the aim of the study was to assess the long-term outcome of an exercise training on PA.  

Statistical analysis

The meta-analysis is performed on very few studies generally, as few as one study from the 1-year follow up in Figure 4, but on average only 2-3 study results were included (Figures 2-4). This could be discussed as a limitation of the meta-analysis.

We thank the reviewer for his/her comment, and we have added this point in the limitations section as suggested: “The meta-analyses are performed on few studies, which cause estimates to be more imprecise. This imprecision, added to the observed heterogeneity, limit our confidence in the results”.

As the number of records screened (1839) seems like a manageable number in this type of review, could you explain the reasoning for the number of reviewers who performed the screening (10 for abstract screening and 12 for full text screening)? Can you explain the reasoning behind data extraction performed in groups (5 pairs and one group of 3)?

The purpose of the present review was decided in a workshop we conducted in Barcelona two years ago, with some international colleagues from the field of healthy ageing. Some PhD students and junior researchers attended the workshop and were willing to participate in the review and learn about the steps and methodology of systematic reviews, as in most cases it was the first time to collaborate in one. After receiving a course, we used the abstract screening and full text data extraction as part of their training.   

Full text versions of potentially eligible studies were retrieved and assessed independently by pairs of authors, and there was not a group of three reviewers; we apologize for the mistake, which has been amended in page 4.

What was the mechanism for standardisation of reviewing for so many reviewers? Was there training for each reviewer using the screening tool?

Yes, there was. As mentioned in the previous point, the PhD students and junior researchers attended a 6-hour introductory course and all reviewers received training on the data extraction tool. We all completed a data extraction of the same paper to assess each item’s responses agreement.   

At the end of the data extraction section 2.3, there is a discrepancy between the number of reviewers who checked the accuracy.

We apologize for the mistake and we have amended it.

Search strategy: as it is common practice to have searches within the last 12 months, it appears that an updated search would be needed.

Additionally, the manuscript states the search was up to January 2018, however the Supplementary Table S1 show the searches conducted in October and November 2017

We conducted an update on January 2018 and we didn’t change the date in the Supplementary material, we have now amended this mistake.

We have done a broad update and we have identified at least two studies that could be included, one of which analyses data of the already included LIFE study (Rejesky et al., 2009). We would be pleased to do an update if more time with the journal’s requirements became available. However, we don’t expect much change in the outcomes.

Reviewer 3 Report

This study systematically reviewed and synthesized study results on long-term exercise adherence. The review was conducted based on the PRISMA statement and the review process is clear. The studied topic i.e. long-term adherence to exercise interventions is of worth in the public health field. Below are my comments for the authors to consider for further improving the quality of this review.

Methods

1.       Why did you not search in SPORTDiscus a major database for sports and sports medicine research?

2.       The systematic search was well done but I do know a few papers that they might have missed. Please check if these papers are eligible.

·         Kriska AM, Bayles C, Cauley JA, LaPorte RE, Sandler RB, Pambianco G: A randomized exercise trial in older women: increased activity over two years and the factors associated with compliance Med Sci Sports Exerc, 1986; 18: 557-562.

·         Pereira MA, Kriska AM, Day RD, Cauley JA, LaPorte RE, Kuller LH. A randomized walking trial in postmenopausal women: effects on physical activity and health 10 years later. Arch Intern Med. 1998;158:1695-1701.

Results

3.       Please add a section on PA outcomes. Currently, results were explained based on self-report and objective measures but it is not very clear how these were defined. Recall periods in the subjective PA measures may be worth noting.

4.       Line 256. “95% CI -0.01 to 0.37” Is this significant? What is your significance level?

5.       Supplementary table S3. “Time point measures”. Hauer et al. 2003. Does “6 and24 months” mean 3 and 21 month post-intervention? Hauer 2003 is included in the 6-12 month follow-up analysis (Figure 2). The same question exists for McMahon et al., 2017. Please clarify how you defined the number of months throughout the manuscript.

Discussion

6.       I agree with the comment on the end-user involvement but I suspect some studies do receive some feedback before the initiation of the trial but did not report that specifically. Do you think this process should be done more systematically and clearly reported?

Author Response

REVIEWER 3

Dear reviewer,

Thank you very much for the time taken to review our systematic review. Please see our answers to your comments in italics, and the changes in the manuscript underlined in yellow. Don’t hesitate to contact us for further information.

Methods

Why did you not search in SPORTDiscus a major database for sports and sports medicine research?

We appreciate your comment and we agree that SPORTDiscus is an important database for sports-related research. Unfortunately, we didn’t have access to this database. However, as SPORTDiscus is hosted on EBSCO, you search multiple databases at the same time. Thus, we believed that using MEDLINE database (also hosted on EBSCO) would cover almost all of the references from SPORTDiscus.

The systematic search was well done but I do know a few papers that they might have missed. Please check if these papers are eligible:

Kriska AM, Bayles C, Cauley JA, LaPorte RE, Sandler RB, Pambianco G: A randomized exercise trial in older women: increased activity over two years and the factors associated with compliance Med Sci Sports Exerc, 1986; 18: 557-562.

Pereira MA, Kriska AM, Day RD, Cauley JA, LaPorte RE, Kuller LH. A randomized walking trial in postmenopausal women: effects on physical activity and health 10 years later. Arch Intern Med. 1998;158:1695-1701.

The authors thank you for your comment. Both papers reported results from the same walking intervention in post-menopausal women. One of the author’s eligibility criteria was to include women aged 50 to 65 years. One of our inclusion criteria was to include older adults aged 65 years or older, so that both papers should had been excluded in the abstract/ full-text screening.

Results

Please add a section on PA outcomes. Currently, results were explained based on self-report and objective measures but it is not very clear how these were defined. Recall periods in the subjective PA measures may be worth noting.

We thank the reviewer for this comment. We have added more information about the self-report measures in Supplementary Table S3, and had added a sentence in section 2.4.

Line 256. “95% CI -0.01 to 0.37” Is this significant? What is your significance level?

Thank you, we’ve deleted the term significant from the text, as the differences observed were only significant for the 6 month follow up.

Supplementary table S3. “Time point measures”. Hauer et al. 2003. Does “6 and 24 months” mean 3 and 21 month post-intervention? Hauer 2003 is included in the 6-12 month follow-up analysis (Figure 2). The same question exists for McMahon et al., 2017. Please clarify how you defined the number of months throughout the manuscript.

We appreciate your comment and have addressed this issue accordingly:

Thanks for pointing out these discrepancies. As you point out, Hauer 2003 data corresponds to 3 and 21 month post-intervention; the 3 month data had been included exceptionally in the 6-11 month category, and this has been indicated in the text.

The study McMahon et al., 2017 reports data at 6 months post-intervention (that’s 8 months from randomization), we have corrected the time point description in Supplementary table S3.

Discussion

I agree with the comment on the end-user involvement but I suspect some studies do receive some feedback before the initiation of the trial but did not report that specifically. Do you think this process should be done more systematically and clearly reported?

The authors believe that participant/patient involvement in the design of any intervention aimed at changing behaviours is important for its success and sustainability. Patient and public involvement (PPI) such as co-creation is hypothesised to have a strong and enduring impact on health outcomes and may be a promising strategy to address other complex health behaviours. However, a clear protocol on how to apply PPI in designing interventions is still lacking, and might be of great interest. Thus, reporting any form of PPI might help standardize such process and might help future intervention design, and we are aware that some studies might have used some form of PPI without reporting it.

Round 2

Reviewer 2 Report

Thank you for addressing the comments and taking the time to amend your manuscript. Unfortunately the methods and content in this manuscript remain fundamentally flawed in my opinion. All of these studies do not measure “physical activity adherence.” In most cases, there is no physical activity level recommendation from a health care provider that the participants are asked to adhere to; therefore this review and analysis is not examining adherence. Additionally, there is no clarity between physical activity and exercise, which are often used synonymously in this manuscript, yet are distinct entities. Many of the strategies discussed were aimed towards the exercise intervention, not physical activity. One example, in section 3.3 Reported strategies to enhance long-term sustainable adherence to PA, the strategy listed in lines 215-217 is a strategy used to enhance exercise progression during the physical therapist led exercise intervention; this is not a strategy used to enhance long-term adherence to a specific recommendation of PA. As PA adherence is the crux and aim of this review, the content does not reflect this. Moreover, the inclusion of Hauer 2003 appears to deviate from the inclusion/exclusion protocol as the intervention took place in an inpatient setting and some participants were discharged to a nursing home, therefore not community-dwelling. I believe that a fully up-to-date, more narrowly focused and clearly defined review of PA adherence OR promotion in older adults would be beneficial to this research field, but unfortunately this review and meta-analysis is not accurately addressing a specific question.

Author Response

Thank you very much for your comments. Please, find below the answers to your comments in italics, and the changes in the manuscript using the track changes function.

1.       There is no physical activity level recommendation from a health care provider that the participants are asked to adhere to, therefore this review and analysis is not examining adherence.

The definition of adherence suggested by the adherence project of the World Health Organization in 2003, was: “the extent to which a person’s behaviour – taking medication, following a diet, and/or executing lifestyle changes, corresponds with agreed recommendations from a health care provider”. However, a recent review stated that there is confusion in the literature about the definition of adherence, and concludes that in the absence of an agreed consensus, the following clear definitions should be used: (a) Health outcomes: completion (i.e., retention), attendance, duration and intensity adherence; (b) Group cohesion/motivation: completion (i.e., retention) and attendance, and (c) Financial viability: attendance (Hawley-Hague et al., 2016).

We agree that if we use the aforementioned definition, we are not assessing adherence but sustainability (maintenance) to physical activity (which includes exercise). We have modified the term throughout the manuscript.

Hawley-Hague HHorne MSkelton DATodd C. Review of how we should define (and measure) adherence in studies examining older adults' participation in exercise classes. BMJ Open. 2016; 6(6): e011560. doi: 10.1136/bmjopen-2016-011560.

2.       There is no clarity between physical activity and exercise, which are often used synonymously in this manuscript.

The authors appreciate you comment and had clarified both terms in the manuscript. We wanted to include exercise-based interventions to enhance the maintenance of any type of physical activity (leisure, house chores/domestic, transport and work-related).

3.       Many of the strategies discussed were aimed towards the exercise intervention, not physical activity.

The authors appreciate you comment and had tried clarify it in the manuscript.

4.       The inclusion of Hauer 2003 appears to deviate from the inclusion/exclusion protocol as the intervention took place in an inpatient setting and some participants were discharged to a nursing home, therefore not community-dwelling.

The authors had an internal discussion about the inclusion of this study. Even though the setting where the intervention took place was an inpatient setting, the majority of participants were community-dwelling. Therefore, we decided to include it.